# Cornelian Cherry (*Cornus mas* L.) Fruit Extract Lowers SREBP-1c and C/EBPα in Liver and Alters Various PPAR-α, PPAR-γ, LXR-α Target Genes in Cholesterol-Rich Diet Rabbit Model

**DOI:** 10.3390/ijms25021199

**Published:** 2024-01-18

**Authors:** Maciej Danielewski, Andrzej Rapak, Angelika Kruszyńska, Małgorzata Małodobra-Mazur, Paweł Oleszkiewicz, Stanisław Dzimira, Alicja Z. Kucharska, Wojciech Słupski, Agnieszka Matuszewska, Beata Nowak, Adam Szeląg, Narcyz Piórecki, Urszula Zaleska-Dorobisz, Tomasz Sozański

**Affiliations:** 1Department of Pharmacology, Wroclaw Medical University, J. Mikulicza-Radeckiego 2, 50-345 Wroclaw, Poland; wojciech.slupski@umw.edu.pl (W.S.); agnieszka.matuszewska@umw.edu.pl (A.M.); beata.nowak@umw.edu.pl (B.N.); adam.szelag@umw.edu.pl (A.S.); 2Hirszfeld Institute of Immunology and Experimental Therapy, Polish Academy of Sciences, R. Weigla 12, 53-114 Wroclaw, Poland; andrzej.rapak@hirszfeld.pl (A.R.); angelika.kruszynska@hirszfeld.pl (A.K.); 3Department of Forensic Medicine, Division of Molecular Techniques, Wroclaw Medical University, M. Sklodowskiej-Curie 52, 50-369 Wroclaw, Poland; malgorzata.malodobra-mazur@umw.edu.pl; 4Department of Radiology and Imaging Diagnostics II, Lower Silesian Center of Oncology, Pulmonology and Hematology, Grabiszynska 105, 53-439 Wroclaw, Poland; pawel@oleszkiewicz.eu; 5Department of Pathology, Wroclaw University of Environmental and Life Sciences, C. K. Norwida 31, 50-375 Wroclaw, Poland; stanislaw.dzimira@upwr.edu.pl; 6Department of Fruit, Vegetable, and Plant Nutraceutical Technology, Wroclaw University of Environmental and Life Sciences, J. Chelmonskiego 37, 51-630 Wroclaw, Poland; alicja.kucharska@upwr.edu.pl; 7Bolestraszyce Arboretum and Institute of Physiography, Bolestraszyce 130, 37-722 Wyszatyce, Poland; npiorecki@ur.edu.pl; 8Institute of Physical Culture Sciences, Medical College, University of Rzeszow, Cicha 2A, 35-326 Rzeszow, Poland; 9Department of General and Pediatric Radiology, Wroclaw Medical University, M. Sklodowskiej-Curie 50/52, 50-369 Wroclaw, Poland; urszula.zaleska-dorobisz@umw.edu.pl; 10Department of Preclinical Sciences, Pharmacology and Medical Diagnostics, Faculty of Medicine, Wroclaw University of Science and Technology, Wybrzeze Wyspianskiego 27, 50-370 Wroclaw, Poland; tomasz.sozanski@pwr.edu.pl

**Keywords:** cornelian cherry, *Cornus mas* L., iridoid, anthocyanin, transcription factor, ATP-binding cassette transporter, adipokine, common carotid artery, atherosclerosis, cardiovascular disease

## Abstract

Cornelian cherry (*Cornus mas* L.) fruits, abundant in iridoids and anthocyanins, are natural products with proven beneficial impacts on the functions of the cardiovascular system and the liver. This study aims to assess and compare whether and to what extent two different doses of resin-purified cornelian cherry extract (10 mg/kg b.w. or 50 mg/kg b.w.) applied in a cholesterol-rich diet rabbit model affect the levels of sterol regulatory element-binding protein 1c (SREBP-1c) and CCAAT/enhancer binding protein α (C/EBPα), and various liver X receptor-α (LXR-α), peroxisome proliferator-activated receptor-α (PPAR-α), and peroxisome proliferator-activated receptor-γ (PPAR-γ) target genes. Moreover, the aim is to evaluate the resistive index (RI) of common carotid arteries (CCAs) and aortas, and histopathological changes in CCAs. For this purpose, the levels of SREBP-1c, C/EBPα, ATP-binding cassette transporter A1 (ABCA1), ATP-binding cassette transporter G1 (ABCG1), fatty acid synthase (FAS), endothelial lipase (LIPG), carnitine palmitoyltransferase 1A (CPT1A), and adiponectin receptor 2 (AdipoR2) in liver tissue were measured. Also, the levels of lipoprotein lipase (LPL), visceral adipose tissue-derived serine protease inhibitor (Vaspin), and retinol-binding protein 4 (RBP4) in visceral adipose tissue were measured. The RI of CCAs and aortas, and histopathological changes in CCAs, were indicated. The oral administration of the cornelian cherry extract decreased the SREBP-1c and C/EBPα in both doses. The dose of 10 mg/kg b.w. increased ABCA1 and decreased FAS, CPT1A, and RBP4, and the dose of 50 mg/kg b.w. enhanced ABCG1 and AdipoR2. Mitigations in atheromatous changes in rabbits’ CCAs were also observed. The obtained outcomes were compared to the results of our previous works. The beneficial results confirm that cornelian cherry fruit extract may constitute a potentially effective product in the prevention and treatment of obesity-related disorders.

## 1. Introduction

Proper nutrition directly impacts the health of the human body, including its development and metabolism, and the functions of individual organs or systems. An unhealthy Western diet, abundant in large amounts of highly processed products and rich in saturated fats and sugar, may contribute to the development of medical conditions known as civilization diseases, such as atherosclerosis, obesity, type II diabetes, or metabolic syndrome [1,2,3]. The newest findings indicate that there are about 600 million people with obesity worldwide, and this figure is projected to increase further still [4]. Obesity is commonly associated with diabetes; the term diabesity has even been established in the literature [5]. In 2021, approximately 529 million people were living with diabetes worldwide, and by 2050, more than 1.31 billion people are forecasted to have diabetes [6]. Diabetes and obesity are commonly linked with cardiovascular disease (CVD), which is the leading cause of death worldwide, with almost twofold more cases than cancer [7]. At the same time, global nutrition trends emphasize increasing the percentage of natural products in the diet. Plant-based or vegan diets, and even healthy foods “to go”, are becoming more and more trendy. Moreover, the common awareness that proper nutrition improves personal well-being, from the gut microbiome to mindfulness, is also increasing [8].

Many natural products, in addition to appropriate nutritional values, can also positively affect health due to the contents of substances with antioxidant, anti-inflammatory, or metabolism-regulatory activities. One of the products with proven beneficial impacts on the functions of the cardiovascular system and the liver is the cornelian cherry fruit (*Cornus mas* L.) [9,10,11,12]. Cornelian cherry fruits are rich in compounds from the iridoid and anthocyanin groups and also contain certain amounts of flavonols, phenolic acids, terpenoids and organic acids [13,14,15].

In our earlier works, we proved that the application of cornelian cherry extract (10 mg/kg of body weight or 50 mg/kg of body weight) has a wide and advantageous effect on various parameters in a cholesterol-rich diet rabbit model [16,17]. We confirmed the beneficial effect of the extract, e.g., on certain transcription factors, adipokines, serum lipids, as well as markers of inflammation, oxidation, and adhesion. Continuing our research to learn the broadest possible characteristics of the cornelian cherry, we decided to evaluate the impact of the extract on markers directly related to, dependent on, or acting as an executive factor for some previously assessed parameters with positive results.

Transcription factors (TFs) are a specific group of proteins and nuclear receptors that affect transcription and are indirectly responsible for the control of protein synthesis. The entire DNA strand never undergoes the transcription process, but only the fragment that encodes a particular protein for which there is a demand in the body. This phenomenon is determined by, among others, transcription factors [18,19]. Five of the TFs play a crucial role in lipid and cholesterol homeostasis: liver X receptor-α (LXR-α), peroxisome proliferator-activated receptor-α (PPAR-α), peroxisome proliferator-activated receptor-γ (PPAR-γ), CCAAT/enhancer binding protein α (C/EBPα) and sterol regulatory element-binding protein 1c (SREBP-1c) [20]. As in the previous work, we observed a significant effect of the application of cornelian cherry extract on LXR-α, PPAR-α, and PPAR-γ expression, and taking into account the fact that often the functions of the TFs are mutually dependent, in this study, we examined the impact of the extract on the other two factors, i.e., SREBP-1c and C/EBPα.

Additionally, amongst the first direct LXR target genes identified were ATP-binding cassette transporter A1 (ABCA1) and G1 (AGCG1)—proteins considered to be essential in lipid metabolism and cholesterol reverse transport [21]. Since we noted a relevant increase in the expression of LXR-α in the liver in both extract groups already, the logical consequence as the next stage of our experiment seemed to be the determination of the ABCA1 and ABCG1 levels.

Lipoprotein lipase (LPL), endothelial lipase (LIPG), fatty acid synthase (FAS), and carnitine palmitoyltransferase 1A (CPT1A) are some of the most important enzymes involved in various stages of lipid and lipoprotein metabolism. While LIPG reduces the plasma high-density lipoprotein (HDL-C) concentration and changes its properties [22], LPL catalyzes the hydrolysis of intravascular triglycerides (TG) packaged in lipoproteins such as chylomicrons (CM) and very-low-density lipoproteins (VLDL-C) into fatty acids (FA) [23]. In turn, cellular fatty acid biosynthesis is conducted by FAS [24], whereas CPT1A constitutes the rate-limiting enzyme for long-chain fatty acid beta-oxidation in the liver [25,26]. Importantly, the function of all these enzymes depends directly or indirectly on the activity of transcription factors whose expression changes under the impacts of extracts confirmed earlier; that is, LXR-α, PPAR-α, and PPAR-γ [27,28,29,30,31,32,33,34,35]. In this report, we showed how the use of the extract affected the concentrations of these enzymes.

In one of our previous works [16], we observed a beneficial impact of cornelian cherry extract on levels of adipokines—an enhancement in adiponectin and diminution in leptin and resistin serum concentrations. Inspired by this outcome, we decided to assess the levels of other adipokines, which depend on the activity of the mentioned transcription factors or previously determined adipokines, such as visceral adipose tissue-derived serine protease inhibitor (vaspin, serpin A12) and retinol-binding protein 4 (RBP4). Vaspin plays a protective role in obesity via its insulin-sensitizing and anti-inflammatory effects [36,37], and RBP4 is contrarily involved in the development of insulin resistance and associated with total cholesterol and triglyceride levels, contributing to the progress of obesity [38,39]. Moreover, we evaluated the expression of one of the adiponectin receptors, AdipoR2. AdipoR2 mainly recognizes full-length adiponectin and is predominantly expressed in the liver, mediating its anti-fibrotic effects [40,41,42].

As we observed a reduction in the thickness of the thoracic and abdominal aorta walls and a diminution in the intima/media (I/M) ratio as an effect of the extract’s application [16], we decided also to determine a resistive index (RI) of common carotid arteries (CCAs) and aortas and perform histopathological evaluations of the CCAs of the rabbits. RI is a pulsatility parameter and describes vascular wall resistance [43].

This study aims to assess and compare whether and to what extent two different doses of resin-purified cornelian cherry extract (10 mg/kg b.w. or 50 mg/kg b.w.), rich in iridoids and anthocyanins, applied in a cholesterol-rich diet rabbit model have an effect on the levels of ABCA1, ABCG1, C/EBPα, SREBP-1c, FAS, LIPG, CPT1A, and AdipoR2 in liver tissue; LPL, vaspin, and RBP4 in visceral adipose tissue; the resistive index of CCAs and aortas; and histopathological changes in common carotid arteries. This article is an attempt to expand the available knowledge, as well as to combine the data obtained with those described in our previous publications so that they constitute a broad, logical, and meaningful step towards the full determination of therapeutic properties of *Cornus mas* L. fruits and their potential usage in the prophylaxis and treatment of cardiovascular diseases (CVDs).

## 2. Results

We have studied the effects of the oral administration of resin-purified cornelian cherry extract on various markers in a cholesterol-rich diet rabbit model.

### 2.1. Quantification of ABCA1, ABCG1, C/EBPα, FAS, LIPG, LPL, SREBP-1c and Vaspin by Enzyme-Linked Immunosorbent Assay (ELISA)

In the assessment of ATP-binding cassette proteins A1 and G1 (Table 1, Figure 1a,b), feeding a cholesterol-rich diet caused a significant decrease in the CHOL groups compared to the P groups. However, the application of cornelian cherry extract led to a substantial increase in EXT10 (*p* = 0.002) in ABCA1 and EXT50 (*p* < 0.001) in the ABCG1 case, compared to the CHOL groups. Importantly, in every case of comparing any EXT group with the CHOL group, the result in the EXT group was always favorable, i.e., higher than in the CHOL group.

Positive, relevant decreases were noted in the assays of SREBP-1c and C/EBPα transcription factors (Table 1, Figure 1c,d). In both cases, the *p* values obtained for both extract groups differed significantly compared to the P and CHOL groups. In the quantification of SREBP-1c, we recorded *p* = 0.002 in EXT10 and *p* < 0.001 in EXT50 vs. P and *p* < 0.001 in EXT10 and EXT50 vs. CHOL. We noted *p* < 0.001 in EXT10 and EXT50 compared to the P and CHOL groups in the C/EBPα assessment.

Moreover, in the EXT10 group as well as the EXT50 group, there was a significant decrease in the level of FAS vs. P group (*p* < 0.001 and *p* = 0.009, respectively). However, when compared to the CHOL, a significant reduction was observed only in EXT10 (*p* = 0.034). The result in the EXT50 group was still lower than in the CHOL group, but the difference in levels was not as noticeable (Table 1, Figure 1e).

As for lipases, we did not observe any meaningful alterations for both LIPG (Table 1, Figure 1f), measured in the liver, and LPL (Table 2, Figure 2a), measured in visceral adipose tissue. However, it is worth mentioning that in the case of LPL, we noted some reduction in the SIMV5 group compared to both groups receiving the extract, where a slight increase vs. P and CHOL groups was recorded. In another assay performed in adipose tissue—the vaspin level measurement (Table 2, Figure 2b)—we did not observe any statistically significant changes.

### 2.2. Assessment of AdipoR2, CPT1A, and RBP4 Expression by Western Blot

The Western blot method was used to determine the levels of the adiponectin receptor 2 (Table 3, Figure 3a) as well as the carnitine palmitoyltransferase 1A (Table 3, Figure 3b) in liver homogenates. Interestingly, there were relevant decreases in the EXT10 group compared to the P group for both markers. However, when comparing to the CHOL group, different results were obtained for the groups receiving the extract, i.e., a significant decrease for the EXT10 according to the CPT1A assessment (*p* = 0.041), but in the case of AdipoR2, a substantial increase in the level (*p* = 0.039) was observed in the EXT50. It is also worth noting that the outcomes in the SIMV5 group were substantially lower as regards the assessed markers, in comparison to both the P and CHOL groups, as well as to both EXT groups.

The Western blot method was also used to determine the levels of retinol-binding protein 4 (Table 3, Figure 3c) in visceral adipose tissue homogenates. We observed a significant decrease in the EXT10 group vs. the P and CHOL groups (*p* = 0.005 and *p* = 0.018, respectively). In other research groups, the fluctuations were relatively small.

### 2.3. Determination of Common Carotid Arteries and Aortas Resistive Index by Ultrasonography

Ultrasonography was used to determine the resistive index in common carotid arteries and aortas of the rabbits. Although we observed an increase in RI in the CHOL vs. P test and a reduction in RI in the EXT50 vs. CHOL test under the CCAs assessment, none of the changes were statistically significant. Generally, the RI values obtained in the CHOL, EXT10, and SIMV5 groups were almost identical (Table 4). We noticed a similar result, i.e., an enhancement in RI in the CHOL vs. P and a reduction in RI, especially in EXT50 vs. CHOL, in the aortic RI assessment. The changes were not statistically significant again; the only one that was was SIMV5 vs. P and CHOL (*p* = 0.020 and *p* = 0.003, respectively). Nonetheless, it is worth noting that in the case of the aorta, the effect of a dose of 10 mg/kg b.w. was more noticeable than in the CCAs case (Table 4).

### 2.4. Histopathological Evaluation of the Common Carotid Arteries

In the histopathological examination of the common carotid arteries, substantial beneficial changes were observed in all groups fed with the assessed compounds, compared to the CHOL group, in which a considerable increase in the occurrence of pathological changes was observed compared to the P group. In both groups receiving the extract and the positive control with simvastatin, there was a significant decrease in unfavorable changes (*p* < 0.001 vs. CHOL in all three cases) and, despite the application of a cholesterol-enriched diet, we saw a reduction to values comparable to the control group. The mean values for the EXT10, EXT50, and SIMV5 groups were 0.700, 0.556, and 0.222, respectively, and ranged between 0 and 1, i.e., the range of no changes to slight damage of endothelial cells, with no signs of atherosclerotic plaque (Table 5). Cross-sections of the common carotid arteries segments with the most acute changes observed in the given groups, stained with hematoxylin-eosin, are depicted in Figure 4.

## 3. Discussion

In our experiment, we evaluated the impacts of two doses (10 mg/kg b.w. or 50 mg/kg b.w.) of resin-purified cornelian cherry extract, abundant in iridoids and anthocyanins, on selected parameters in liver and visceral adipose tissue in a cholesterol-rich diet rabbit model. In addition, the resistive index of the common carotid arteries and aortas was determined, and the histopathological evaluation of the CCAs was performed.

In the study of the SREBP-1c level, we obtained a significant decrease in both extract groups, compared to the control group and the group fed only a diet enriched with 1% cholesterol. This is a beneficial result because sterol regulatory element-binding protein-1c is a crucial transcription factor regulating genes essential for the biosynthesis and uptake of lipids. It activates the synthesis of fatty acids and triglycerides, causing the accumulation of lipids [44,45]. SREBP-1c overexpression also leads to an increase in VLDL-C and a decrease in HDL-C [46,47].

SREBP-1c is nutritionally regulated [48]. Karasawa et al. demonstrated that the absence of SREBP-1 suppresses Western diet-induced hyperlipidemia in LDLR-deficient mice and ameliorates atherosclerosis [49]. Our results accord with this hypothesis—the increase in the level of SREBP-1c in the CHOL group compared to the P group was not statistically significant, but was noticeable. On the other hand, the oral administration of the extract substantially reduced the level of SREBP-1c expression in the liver.

To date, research on the effect of products of the *Cornus* genus on SREBP-1 has been conducted mainly by Park et al. In the case of morroniside, an iridoid compound isolated from Corni Fructus, the researchers obtained results consistent with ours, i.e., a drop in the level of SREBP-1 in the liver [50]. Similar results were also achieved with another iridoid, loganin [51], the polyphenolic compound 7-O-galloyl-D-sedoheptulose [52], and the total triterpenoid acids fraction [53]. However, no significant changes in SREBP-1 expression were observed in the case of the oral administration of the fruit extract in the rat model [54]. Importantly, all studies were conducted on derivatives of *Corni Fructus*, i.e., *Cornus officinalis*, but none of them concerned *Cornus mas* L.

As in the case of SREBP-1c, the use of cornelian cherry extract resulted in a significant decrease in the expression of C/EBPα in the liver, compared to the P and CHOL groups. This is also a desirable outcome because CCAAT/enhancer binding protein α is pivotal in adipogenesis (regulates final stages of adipocyte differentiation), lipogenesis, and glucose metabolism, among others in the liver [37,55]. Similar results were obtained in other *Cornus* studies [56,57,58], but again, none of them concerned *Cornus mas* L. Factors from the C/EBP group—a family of six, ranging from C/EBPα to C/EBPζ—have been proven to play an important role in regulating genes involved in a variety of physiological processes ranging from glucose, lipid, and energy homeostasis to the production of adipokines, acute-phase response proteins and hemostatic factors, and the development of type II diabetes and CVD [58,59]. Moreover, an enhancement in C/EBPα has been reported to transcriptionally activate a boost in the expression of obese (ob) genes in the mouse model [60].

Both of the assessed transcription factors, along with three others—PPAR-α, PPAR-γ, and LXR-α—play a key role in the regulation of lipid and cholesterol homeostasis. In our previous work [16], we observed favorable effects related to the extract’s application regarding the other three mentioned transcription factors. We confirmed a significant increase in the expression of LXR-α in the liver in both EXT groups, as well as a significant increase in PPAR-α in the EXT10 group and PPAR-γ in the EXT50 group in the aorta. To summarize, the use of the cornelian cherry extract can substantially and positively affect the expression of all major transcription factors involved in cholesterol and lipid metabolism. It is worth emphasizing that, except PPARs, this impact is relevant regardless of the dose of the extract used. The influence on transcription factors is currently one of the most ubiquitous mechanisms of action of drugs commonly used in the treatment of atherosclerosis, e.g., fibrates or colesevelam. Therefore, it seems legitimate, supported by the results obtained, to conclude that due to the significantly beneficial effects on transcription factor expression, *Cornus mas* L. fruit extract may represent an effective therapeutic agent and a valuable alternative to the currently used anti-atherosclerotic therapies; ergo, proper human trials are more than recommended.

It is important to state, though, that all of these transcription factors do not work in parallel, independently of each other, but are linked by a network of mutual dependencies. For example, the upregulation of SREBP-1c may be mediated by LXR-α activation [61,62], which in turn may be mediated by PPAR-γ augmentation [63]. SREBP-1c is a target of LXR-α because the promoter of the SREBP-1c gene contains an LXR reaction element (LXRE) [64]. In addition, both C/EBPα and PPAR-γ can induce each other’s expression via a positive feedback loop [65]. This means that, when omitting other parameters and focusing on the extract’s effect on TFs, the beneficial effects achieved are not the result of the sum of positive changes in the expression of individual transcription factors, but rather the effect of their complex, multifaceted action.

In the assessment of ATP-binding cassette transporters, we also obtained favorable results. A cholesterol-rich diet caused relevant decreases in the CHOL group compared to the P group, while the use of the extract contributed to an enhancement in the expression levels of both proteins, with a significant change in ABCA1 in the EXT10 group and ABCG1 in the EXT50 group. In both cases, this increase led to the acquirement of transporters’ expression at values comparable to group P. This outcome is consistent with the work of Gao et al. on morroniside isolated from *Cornus officinalis* [66].

Cholesterol in cells is used as a component of the cellular membrane or a precursor of steroid hormones. Excess cholesterol in cells is excreted by ATP-binding cassette transporters. ATP-binding cassette proteins A1 and G1 are crucial proteins that maintain cholesterol homeostasis [67,68]. One of their primary functions is the efflux of intracellular excess cholesterol and phospholipids across the plasma membrane to combine with mainly apolipoprotein A1 and form nascent HDL-C, which constitutes the first step of reverse cholesterol transport [69,70,71]. Recent studies on atherosclerosis pathogenesis show that isolated monocytic cells subjected to primary stimulation with oxLDL-C (modified, oxidized LDL-C), during secondary stimulation with agonists of toll-like receptors TLR2 and TLR4, exhibit an increased production of pro-inflammatory cytokines such as TNF-alpha, interleukins 6, 8 and 18, and the MCP-1 chemokine. An oxLDL-C cell stimulation results in an enhancement in the expression of CD36 and SRA (scavenger class A) receptors responsible for oxLDL-C uptake, which, in conjunction with the also-observed drop in the reverse cholesterol transporters ABCA1 and ABCG1, contributes to a meaningful accumulation of lipoproteins in macrophages. The increased uptake of oxLDL-C by CD36 and SRA and the restricted efflux of cholesterol result in the macrophages’ transformation into foam cells [62,63,64,65,66,67,68,69,70,71,72,73,74].

ABCA1 and ABCG1 are induced by the PPAR-γ-LXR-α-ABCA1/ABCG1 pathway [63,67,75,76,77]. LXR-α activation leads to the robust upregulation of ABCA1 and ABCG1 in macrophages, the intestine, and the liver [21]. Thus, our results form an integral whole—the previously confirmed significant augmentation in the expression of PPAR-γ in EXT50 and LXR-α in both groups receiving the extract translates into a positive effect of an increase in the reverse cholesterol transporters levels. However, this is not the first outcome in which we observe different levels of efficacy in the response of individual parameters to two doses of the extract. This type of result confirms the extract’s effectiveness in general, but at the same time, it indicates that in the optimization of the prospective extract’s usage in prophylaxis or therapy, it will be necessary to select the appropriate dose carefully. It is also a matter of consideration whether, in the case of a human trial, the number of doses used in the experiment should be increased to understand better the potential dose-dependence of the extract’s effect on the studied markers.

For fatty acids to enter the cell, circulating lipoproteins must be hydrolyzed. Extracellular, water-soluble enzymes called lipases bind to lipoproteins and catalyze their hydrolysis, releasing their contents [78]. Lipases hydrolyze water-insoluble lipid molecules, such as triglycerides, phospholipids, and galactolipids. They are ubiquitous in nature and are present in almost every living organism [79]. The triglyceride lipase gene family plays a critical role in lipoprotein metabolism and includes the lipases lipoprotein lipase (LPL), hepatic lipase (HL), and endothelial lipase (LIPG) [78]. In the present study, we determined the concentrations of lipoprotein lipase in visceral adipose tissue and endothelial lipase in the liver. We obtained only minor favorable outcomes without statistical significance. In the groups receiving the extract, we noted a slight enhancement in LPL and a slight depletion in LIPG levels. Thus, the previously reported substantially augmented expression of LXR-α, PPAR-α, and PPAR-γ did not translate into relevant alterations in these target compounds. Although the changes were modest in both cases, there was another noticeable differentiation between the extract’s dose and the effect on the analyzed parameter, i.e., the increase in LPL was greater in EXT50, while we saw a decrease in LIPG in the EXT10 group.

Lipoprotein lipase is highly expressed in tissues that oxidize or store fatty acids in large quantities, such as the heart, skeletal muscle, and adipose tissue [80]. It is involved in the metabolism of all classes of lipoproteins, including the clearance of chylomicron remnants, the formation of intermediate-density lipoproteins and low-density lipoproteins, and the catabolism of high-density lipoproteins [81,82]. LPL deficiency triggers severe hypertriglyceridemia [83,84], whereas its overexpression may lead to insulin resistance and obesity [85,86]. In turn, endothelial lipase is synthesized and expressed in the endothelium of the majority of tissues. The degree of expression is higher in richly vascularized tissues than in the less vascularized ones. This enzyme is considered the main, inversely correlated regulator of the plasma HDL-C concentration. It has been demonstrated to be upregulated in inflammatory conditions such as atherosclerosis and cardiovascular disease [22,87,88].

There are not many data regarding the effects of *Cornus* genus products on the featured lipases. In 2018, Khan et al. reported that the anthocyanins-rich fraction of the ethanolic leaf extract of *Cornus kousa* downregulated LPL levels (an opposite trend to that presented in our results) [57], while our description of the *Cornus*-LIPG interaction is, to our knowledge, the first report on this subject to date. Therefore, much more research is needed before appropriate conclusions can be drawn.

In the evaluation of the level of fatty acid synthase in the liver, we observed the greater effectiveness of the lower dose of the extract—a significant reduction compared to both the P and CHOL groups, while in a higher dose, the reduction was only significant compared to the P group. The slightly lower level of FAS in the CHOL compared to the P may be due to the fact that dietary fats inhibit FAS expression to decrease de novo lipogenesis when fats are already abundant, which is relatively rapidly obtainable in the CHOL group case.

FAS is traditionally considered a housekeeping protein, producing fatty acids that can be used for energy storage, membrane assembly and repair, and secretion in the form of lipoprotein triglycerides. However, recent findings suggest that hepatic FAS may also be involved in signaling processes that include the activation of peroxisome proliferator-activated receptor α, which mediates the adaptive response to fasting by promoting the transcription of genes involved in the uptake and catabolism of fatty acids, also thereby reducing FAS activity [89,90]. Moreover, polyunsaturated fatty acids may suppress the FAS promoter by inhibiting SREBP-1c activity [91]. On the other hand, SREBP, activated by insulin, is involved in the nutritional induction of expression of adipogenic genes, including FAS, because the FAS promoter contains a sterol regulatory element [92].

The inhibition of fatty acid synthase was observed in other *Cornus* studies [57,93]. The relatively small decrease in FAS that we observed, despite a relevant increase in PPAR-α and decrease in SREBP-1c, may be because applying the cornelian cherry extract also contributed to an augmentation in LXR-α, which can stimulate FAS directly and indirectly [94], and PPAR-γ, the expression of which shows a positive correlation with FAS activity [95,96].

High-fat diets can contribute significantly to the increase in CPT1A expression in the liver [97]. The results of our experiment do not confirm this statement. The level of CPT1A in CHOL was slightly lower than in P, and EXT50 was higher in CHOL. In the case of a dose of the extract of 10 mg/kg b.w., we noted a significant decrease in CPT1A compared to the CHOL and P, but this decrease was relevantly smaller than in the positive control—the SIMV5 group. Therefore, the obtained results are difficult to interpret unambiguously.

CPT1A is one of the major lipolytic markers [98]. It catalyzes the transfer of the long-chain acyl group in acyl-CoA ester to carnitine, allowing fatty acids to enter the mitochondrial matrix for oxidation [99]. However, the overexpression of CPT1A in the liver, induced by saturated fatty acids and reactive oxygen species, triggers mitochondrial activity and may lead to hepatic steatosis and fibrosis [100,101]. The effects of CPT1A are mainly mediated by peroxisome proliferator-activated receptor α, as CPT1A is one of the PPAR-α target genes [97,102,103]. Multiplicitous evidence implies that PPAR-α contributes to the homeostasis between lipogenesis and lipolysis, ketone body production, and cholesterol metabolism by regulating 3-hydroxy-3-methylglutaryl (HMG)-CoA synthase 2 and malonyl-CoA decarboxylase [104]. That may explain the changes in CPT1A levels not only in the extract groups but also in the simvastatin group.

In light of the above, the results elicited from several of our studies seem very interesting. In the EXT10 and SIMV5 groups we noted a significant enhancement in PPAR-α expression, which, according to numerous reports [102,103,105,106], should translate into an increase in the level of CPT1A. Meanwhile, we observed a negative correlation. A proper account of this outcome is also not facilitated by the fact that we have not found any source describing the effects of *Cornus* derivatives on CPT1A activity. Therefore, we can only hypothesize that the obtained result may have been influenced by the increased expression of hepatic transcription factor of fatty acid anabolism, LXR-α, which was also observed [16,107]. Moreover, the experiment’s relatively short duration might have already contributed to the activation of PPAR-α, but not its target genes yet. However, these are only speculations that require further investigation due to the absence of other research.

In the first assessment to our knowledge, of the vaspin adipokine level in the *Cornus* genus, we observed a slight increase in the CHOL compared to P. Administration of the cornelian cherry extract resulted in a minor depletion in the level, with a greater decrease following the 50 mg/kg b.w. dose. However, none of these changes were statistically significant. Vaspin is suggested as a compensatory molecule in obesity and insulin resistance [108]. Its expression increases with body weight gain [109]. So far, several factors regulating the level of vaspin have been described. In addition to the above-mentioned obesity and insulin resistance, another adipokine—leptin—also showed a positive correlation [110,111]. In turn, vaspin has a normalizing effect on the expression of further adipokines—resistin, adiponectin, and, in a feedback loop, leptin [112]. This is confirmed by the results of our experiments. The previously observed [16] decrease in leptin and resistin and the increase in adiponectin compared to CHOL are reflected in a slightly lower vaspin level in both extract groups.

Moreover, vaspin can induce the expression of PPAR-γ and C/EBPα [113]. In the previous report [16], we noted an increase in PPAR-γ, but a relevant decrease in C/EBPα in the present. It should, however, be remembered that we observed only minor fluctuations in vaspin amongst study groups, so overall, its level presumably had a modest effect on changes in the expressions of these TFs. The literature also attributes significant anti-inflammatory effects to vaspin, including reducing the expression and secretion of pro-inflammatory cytokines and adhesion molecules, such as i.a. IL-6, MCP-1, VCAM-1, and ICAM-1 [114,115,116,117]. In both groups receiving the extract, we had previously observed, in most cases, a decrease in the levels of these markers [17], so it can be hypothesized that vaspin was one of the factors that contributed to such an outcome to some degree, and from this perspective, the lack of a meaningful reduction in the level of vaspin vs. CHOL can be considered a positive effect.

As for the AdipoR2 assessment, we observed a decrease in the expression of this receptor in the liver in the CHOL group compared to the P group. The application of the extract at a dose of 10 mg/kg b.w. resulted in a further depletion, while for EXT50, we noted a significant increase in expression compared to CHOL. We did not find any comparative data, not only regarding the *Cornus* genus but also for compounds from the iridoid or anthocyanin groups, i.e., the main active ingredients of the extract. It is also worth mentioning the substantial drops in AdipoR2 in the SIMV5 group, which may suggest the possible restriction of the statins’ impact on hepatocytes by reducing the effect of adiponectin due to the limited ability to form ligand-receptor complexes. Similar results in the intensive statin therapy model in the human monocyte–macrophage lineage were shown by Gasbarrino et al. [118].

Adiponectin is one of the most important adipokines secreted by adipocytes. It exerts its properties via two receptors, AdipoR1 and AdipoR2. The binding of adiponectin to AdipoR2 in the liver activates AMP-activated protein kinase (AMPK), which stimulates the phosphorylation of acetyl-CoA carboxylase (ACC) and activates PPAR-α and PPAR-γ signaling pathways, leading to a reduction in hepatic lipogenesis, the enhancement of fatty acid oxidation, and the alleviation of hepatic steatosis. In addition to its primary function—improving abnormal glucose and lipid metabolism—adiponectin can also reduce the inflammatory response and oxidative stress in the liver [119,120,121,122]. Alzahrani et al. reported that AdipoR2 KO mice had an enhanced fibrotic signature with increased i.a. MMP-2 and MMP-9 [41]. We found that the 50 mg/kg b.w. dose of the extract in particular mitigates the mRNA expression of MMP-1 and MMP-9 [17], which may be partly related to the noticeable increase in AdipoR2 in the EXT50 group. Moreover, adiponectin, via AdipoR2, may suppress hepatic SREBP-1c expression [123,124] and increase ATP-binding cassette transporter A1 and lipoprotein lipase, resulting in the decrease in TG [125,126,127,128]—all of which, with differentiated degrees of intensity, we observed in this and a previous [16] study.

Hepatokines, including retinol-binding protein 4 (RBP4), are proteins secreted by hepatocytes, many of which are associated with the induction of metabolic dysfunctions [129]. Retinol-binding protein 4 is involved in the development of i.a. insulin resistance [39]. RBP4 transports retinol from the liver to peripheral tissues. Visceral fat accumulation causes inflammation and hormonal adipose tissue dysfunction, which leads to RBP4 overproduction [38]. Excessive levels of RBP4 in the liver and adipose tissue are observed in morbidly obese patients [130]. RBP4 stimulates the expression of adhesion molecules in the endothelial cells, promoting the progress of atherosclerosis and arterial hypertension. The augmentation of RBP4 has been demonstrated as a marker of type 2 diabetes, the severity of atherosclerosis, and the risk of cardiovascular events in population studies [38].

Interestingly, we observed a slight decrease in the level of RBP4 in the CHOL group compared to the P group. In general, the fluctuations in the four study groups, i.e., P, CHOL, EXT50, and SIMV5, were minor, and the results obtained in these groups could suggest that cornelian cherry fruit extract does not affect the occurrence of RBP4 in visceral adipose tissue. However, in the EXT10 group, we recorded substantially lower RBP4 levels compared to all other groups, including the comparison group and CHOL. It seems reasonable to ask whether such a significant difference in results from other research groups may be the consequence of an error made during the analysis. The standard deviation values for the EXT10 group are not worryingly high (lower than, for example, the EXT50 group) and may indicate the relative repeatability of the outcomes for individual samples within the group. Is it possible, then, that the reason for this is the dose of the extract used. Frequently, as the dose increases, the pharmacological effect changes qualitatively, i.e., it may initially stimulate, then irritate, inhibit, and finally paralyze the selected function. Perhaps the effect on RBP4 is uppermost at a dose of 10 mg/kg b.w., and as the dose increases, that impact declines. In the case of other parameters, we have often observed the greater effectiveness of a lower extract dose, but never with such a difference.

Although, to our knowledge, no one has examined the influence of *Cornus* products on RBP4, there are some sources regarding the individual components of the extract. Vendrame et al. confirmed a reduction in RBP4 expression in the rats’ adipose tissue after applying a diet enriched with wild blueberries abounding in anthocyanins [131], while Sasaki et al. obtained a similar result for cyanidin 3-glucoside in a mouse model [132]. Cyanidin and its derivatives are one of the main active ingredients of the cornelian cherry fruit extract tested (Figure 1). In turn, in the case of geniposide—a compound from the iridoid group, which constitutes the second leading group of active ingredients in the extract—disturbances in circulating RBP4 level, including its synthesis, secretion and homeostasis, were observed [133]. However, none of these reports refer to the issue of dose dependence in an unambiguous manner, so it is impossible to perform even a cursory comparative analysis. Therefore, further studies are more than recommended, especially considering the significant decrease in RBP4 we observed in the EXT10 group.

The resistive index, also known as the Pourcelot index, is a calculated measure of pulsatile blood flow that reflects the resistance to blood flow caused by the microvascular bed distal to the site of measurement. It is proportional to not only vascular resistance but also vascular compliance, and increases as the vessel narrows—one of the parameters influencing RI is the intima/media (I/M) ratio [134,135,136]. RI in the common carotid artery compares the velocity of systolic and diastolic blood flow. CCA RI also expresses the stiffness and the vascular resistance of the aorta. Aortic stiffness is associated with vascular impairment and the risk of cardiovascular diseases, e.g., atherosclerosis or hypertension [135,137].

The RI values depend largely on the target organ. The blood vessels supplying vital organs such as the internal carotid, hepatic, renal, and testicular arteries generally have a low RI, with values varying within a range of 0.55–0.7. In turn, blood vessels supplying body extremities, for example, the external carotid, external iliac, axillary, and mesenteric arteries, have a high RI, with values over 0.7 [138]. In our study, regardless of whether it was in the CCA or the aorta, we observed an increase in RI in the CHOL group compared to the P group, and this unfavorable effect was alleviated in the EXT50 group. This outcome is consistent with our earlier aorta’s I/M ratio assessment [16], where the dose of 50 mg/kg b.w. also demonstrated higher efficacy.

Furthermore, the application of cornelian cherry extract protected against the development of atheromatous changes in the common carotid arteries of the rabbits. After 60 days of the experiment, there were no visible atherosclerotic changes in the CCAs of rabbits in the control group. In contrast, the cholesterol-fed rabbits evolved considerable atheromatous changes in the CCAs. The typical atherosclerotic plaque was composed of inflamed cells, proliferated fibroblasts, assembled macrophages and foam cells, fibrin, and amorphous masses. Administering both doses of the extract (with a slightly greater impact of the 50 mg/kg b.w. dose) and simvastatin significantly prevented the formation of atherosclerosis in the CCAs. This is a visible effect of the extract’s influence and constitutes, to a certain extent, a direct summary of its administration, and a physical confirmation of the described impact on the expressions of various genes, as well as enzyme activity, and the function of the cardiovascular system and liver in the context of lipid and cholesterol metabolism.

The research we conducted in several studies concerned diverse parameters, including transcription factors, enzymes, inflammation, adhesion markers and histological and hemodynamic changes, and was carried out in various materials, such as in serum, liver, aortas, and adipose tissue. However, mutual comparisons pertaining to the obtained results are justified because, although the expressions of all determined parameters may differ in level, depending, for example, on ligand concentration or cell type, there is no doubt that overall, a complex network of mutual dependencies, impacts, and feedback loops is formed.

Our experiment has its limits. Of paramount importance, we consider a difficult-to-assess dose–effect relationship, which, given the variable impact obtained, is crucial for the design of subsequent studies. The lack of a “carbohydrate component” of the Western diet in the model and its relatively short duration, sometimes too brief to induce metabolic changes, is also worth mentioning. Nevertheless, our experiment sheds valuable light on the effects of cornelian cherry fruit extract, and the rabbit model stands out among commonly conducted mouse or rat studies.

## 4. Materials and Methods

### 4.1. Animal Model

To perform the current study, we used biobank liver and fat tissue samples collected during the original experiment described in our previous works [16,17]. The animal care and all experimental procedures were in accordance with the applicable international, national, and institutional guidelines and approved by the Local Ethics Committee for Animal Experiments at the Hirszfeld Institute of Immunology and Experimental Therapy of the Polish Academy of Science in Wroclaw (Approval code 21/2015). Fifty sexually mature male New Zealand white rabbits aged 8 to 12 months (body weight range: 2.664–3.635 kgs) were used in the 60-day experiment. The animals were housed in individual chambers with temperatures maintained at 21–23 °C. The rabbits were acclimatized, observed for four weeks, and then randomly divided into five groups of 10 animals. During the study, they had free access to drinking water and received the same daily portion of chow (40 g/kg). The animals in group P were fed the standard chow for rabbits. Animals in other groups (CHOL, EXT10, EXT50, and SIMV5) were fed with the standard chow enriched with 1% cholesterol. For the 60 days of the experiment, the following substances were administered orally, once daily in the morning, to the rabbits: groups P and CHOL—normal saline solution, group EXT10—*Cornus mas* L. extract 10 mg/kg/b.w., group EXT50—*Cornus mas* L. extract 50 mg/kg/b.w., and group SIMV5—simvastatin 5 mg/kg/b.w. as a positive control (Table 6).

At the end of the study, the rabbits were put under terminal anesthesia with the use of Morbital^®^ (Biowet, Puławy, Poland; 1 mL of the drug contains 133.3 mg of sodium pentobarbital and 26.7 mg of pentobarbital) at a dose of 2 mL/kg given intraperitoneally (i.p.) [16]. The samples of livers and fat tissues were afterward harvested and cleaned, then frozen and stored at −70 °C for further analysis.

### 4.2. Plant Materials and Preparation of the Extract

The fruits of cornelian cherry (*Cornus mas* L.) were collected at the Arboretum and the Institute of Physiography in Bolestraszyce, Poland, and stored at −20 °C. The herbarium specimen (BDPA 3967) was authenticated and deposited at the Herbarium of the Arboretum and the Institute of Physiography in Bolestraszyce, Poland.

The research material was resin-purified cornelian cherry fruit extract (EXT) derived according to the recipe described in our previous work [17]. The essential ingredients of the extract, determined by the HPLC-PDA method [139], constituted active compounds from the groups iridoids, anthocyanins, phenolic acids, and flavonols. The qualitative and quantitative compositions of the substances detected in the EXT are presented in Figure 5, and the chemical structures of dominant iridoids and anthocyanins are presented in Figure 6.

### 4.3. Quantification of ABCA1, ABCG1, C/EBPα, FAS, LIPG, LPL, SREBP-1c, and Vaspin by Enzyme-Linked Immunosorbent Assay (ELISA)

The ELISA method was used in the evaluation of liver homogenates (Homogenizer PRO250, PRO Scientific Inc., Oxford, CT, USA), checking for the levels of:ABCA1—Rabbit ATP Binding Cassette Transporter A1 ELISA Kit, ELK9787, ELK Biotechnology Co., Ltd. (Denver, CO, USA);ABCG1—Rabbit ATP Binding Cassette Transporter G1 ELISA Kit, QY-E30356, QAYEE Bio-Technology Co., Ltd. (Shanghai, China);C/EBPα—Rabbit CCAAT/Enhancer Binding Protein Alpha ELISA Kit, ELK9791, ELK Biotechnology Co., Ltd. (Denver, CO, USA);FAS—Rabbit Fatty Acid Synthase ELISA Kit, ELK9790, ELK Biotechnology Co., Ltd. (Denver, CO, USA);LIPG—Rabbit Endothelial Lipase ELISA Kit, ELK9795, ELK Biotechnology Co., Ltd. (Denver, CO, USA);SREBP-1c—Rabbit Sterol Regulatory Element-Binding Protein 1c, ELISA Kit, ELK9798, ELK Biotechnology Co., Ltd. (Denver, CO, USA).

The ELISA method was used in the evaluation of visceral adipose tissue homogenates (MagNA Lyser Instrument, Roche Diagnostics, Rotkreuz, Switzerland) for checking levels of:LPL—Rabbit Lipoprotein Lipase ELISA Kit, ELK9789, ELK Biotechnology Co., Ltd. (Denver, CO, USA);VASPIN—Rabbit Visceral Adipose Specific Serine Protease Inhibitor ELISA Kit, QY-E30362, QAYEE Bio-Technology Co., Ltd. (Shanghai, China).

All tests were performed according to the manufacturer’s instructions. All concentrations were expressed as ng/mL or pg/mL.

### 4.4. Assessment of AdipoR2, CPT1A, and RBP4 Expression by Western Blot

Samples of rabbit livers (for AdipoR2 and CPT1A assay) were homogenized in a buffer containing 25 mM Tris pH 7.5, 50 mM NaCl, 1% NP-40, and a set of protease inhibitors. Samples of rabbit visceral adipose tissue (for RBP4 assay) were homogenized in RIPA buffer containing 50 mM Tris HCl pH 7.4, 150 mM NaCl, 1.0% NP-40, 0.5% Sodium Deoxycholate, 1.0 mM EDTA, 0.1% SDS, 0.01% sodium azide, and a set of protease inhibitors. The samples were centrifuged. The obtained supernatant was mixed with the SDS sample buffer, then boiled at 95 °C for 5 min, and afterward subjected to SDS-PAGE on 12% gel. The cleaved proteins were transferred to the PVDF membrane (Thermo Fisher Scientific, Waltham, MA, USA) using semi-dry transfer. Next, the membrane was blocked overnight with 1% casein in TBS at 4 °C and incubated with proper antibodies.

We applied 1 µg/mL of the antibodies anti-AdipoR2 (ADIPOR2 Antibody, ARP60819_P050, Aviva Systems Biology, Corp., San Diego, CA, USA), anti-CPT1A (CPT1A Antibody, ARP44796_P050, Aviva Systems Biology, Corp., San Diego, CA, USA) and anti-RBP4 (RBP4 Antibody, ARP41578_P050, Aviva Systems Biology, Corp., San Diego, CA, USA) in subsequent markings with beta-actin C-04 (Santa Cruz Biotechnology Inc., Santa Cruz, CA, USA) at room temperature for 1 h, followed by a secondary horseradish peroxidase-labeled antibody (Dako, Glostrup, Denmark). The bounded antibodies were visualized using the West–Pico blotting detection system (Thermo Fisher Scientific, Waltham, MA, USA). The blots were scanned, and the optical density of the bands was analyzed with ImageJ v. 1.53 software [12,16,140].

### 4.5. Determination of Common Carotid Arteries and Aortas Resistive Index by Ultrasonography

The ultrasonography of common carotid arteries and aortas was carried out on the last day (60th) of the study. The examination was performed using a Toshiba Aplio 500 device (Toshiba, Tokyo, Japan) and Micro-Convex 11MC4 (PLT-712BT) 4–11 MHz high-frequency probe (Toshiba, Tokyo, Japan) for CCAs assessment, or a Toshiba PLT-805AT linear probe (Toshiba, Tokyo, Japan) for aortas assessment, by the same radiologist for all subjects. The examined areas of the animals were shaved to the skin. The rabbits were immobilized on the back in the supine position (aided by an assistant). The flow in the vessels was examined at an angle of less than 60°, parallel to the flow stream, and RI was determined. The resistive index was calculated according to the formula:RI = (PSV − EDV)/PSV
where PSV = peak systolic velocity and EDV = end-diastolic velocity.

### 4.6. Histopathological Evaluation of the Common Carotid Arteries

The material fixed in buffered 7% formalin was embedded in paraffin and cut into 4 µm sections, which were stained via the routine hematoxylin–eosin method. The cut sections were mounted on Superfrost Plus slides, deparaffinized in xylene, and rehydrated to distilled water. The samples were incubated in Mayers Hematoxylin for 3 min and washed with 3 changes of tap water or until blue stopped coming off the slides. Next, the slides were counterstained in alcoholic eosin for 10 s without rinsing, dehydrated through 2 repetitions of 95% ethanol, 2 repetitions of 100% ethanol and 2 repetitions of acetone, before being cleared in 2 repetitions of Xylene (10 s each) in each. Finally, the slides were mounted with cover slides [141].

Evaluation was performed by the same pathologist for all subjects. The preparations were viewed without knowing the divisions into the control group and experimental groups. Microscopic analysis was performed using an Olympus BX53 light microscope coupled with an Olympus UC90 camera. The intima and media thickness measurements were made using the cellSens Standard V.1 software (Olympus, Tokyo, Japan). Photos were taken at an enlargement of 200×; the scale shown in the photos is 50 µm.

The changes, depending on their severity, were assigned an appropriate score with the following classification:No changes;Slight damage of endothelial cells, no signs of atherosclerotic plaque;Single foam cells or their small clusters on the endothelial surface;Focally located large plaque/foam cell cluster;Large focus or several individual clusters of foam cells;Very extensive atherosclerotic plaque, often covering the entire circumference of the vessel.

### 4.7. Statistical Analysis

The numeric values obtained were expressed as mean ± standard deviation (mean ± SD). The statistical analysis was conducted using Statistica v. 13.3 software (TIBCO Software, Inc., Palo Alto, CA, USA). One-way analysis of variance (ANOVA) with least significant difference (LSD) Fisher’s post hoc test was performed to compare the experimental groups. The *p*-values < 0.05 were considered statistically significant. Graphical representations of the statistical data were created using the GraphPad Prism v. 10 software (GraphPad Software, Inc., Boston, MA, USA).

## 5. Conclusions

The oral administration of resin-purified *Cornus mas* L. extract rich in iridoids and anthocyanins, in doses of 10 mg/kg b.w. or 50 mg/kg b.w., exhibited a diversified impact on levels of various markers altered by a cholesterol-rich diet in the rabbit model. The positive (lowering) effects of both doses were observed in the case of transcription factors SREBP-1c and C/EBPα. In the assessment of some parameters, we obtained beneficial results only in the case of one dose: EXT10 significantly increased ABCA1 and decreased FAS, CPT1A, and RBP4, and EXT50 significantly enhanced ABCG1 and AdipoR2. Additionally, we noted a relevant mitigation in atheromatous changes in rabbits’ CCAs (both doses) and a positive effect of a higher dose of the extract on RI in the common carotid arteries and aortas, but the reduction in value, although noticeable, was not statistically significant.

Our study, to our knowledge, tested for the first time the influence of *Cornus* derivative on CPT1A, AdipoR2, RBP4, LIPG, and vaspin, with a relevant outcome for the first three. We also compared the results received in this experiment to those described in two previous reports, thus creating a more comprehensive view of the effects of the extract. Considering that we did not observe any side effects during the experiment, cornelian cherry fruit extract may constitute a valuable product to be used in the prevention and treatment of diseases related to the Western diet, such as atherosclerosis or metabolic syndrome.

## Figures and Tables

**Figure 1 ijms-25-01199-f001:**
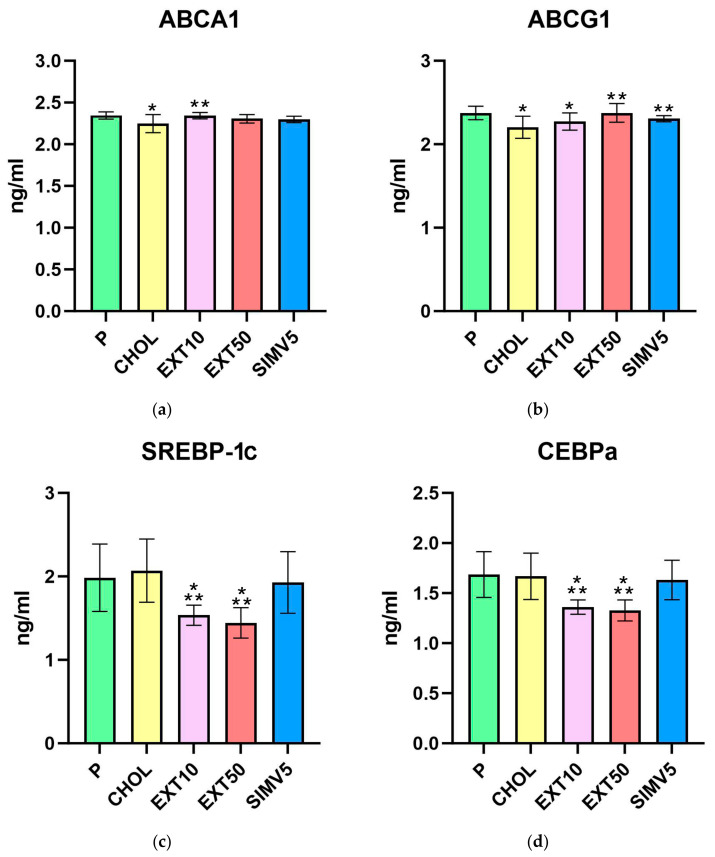
Liver homogenates levels of (**a**) ATP-binding cassette protein A1 (ABCA1), (**b**) ATP-binding cassette protein G1 (ABCG1), (**c**) sterol regulatory element-binding protein-1c (SREBP-1c), (**d**) CCAAT/enhancer binding protein alpha (CEBPa), (**e**) fatty acid synthase (FAS), and (**f**) endothelial lipase (LIPG) by ELISA method. P—standard chow; CHOL—standard chow + 1% cholesterol; EXT10—standard chow + 1% cholesterol + cornelian cherry extract 10 mg/kg b.w.; EXT50—standard chow + 1% cholesterol + cornelian cherry extract 50 mg/kg b.w.; SIMV5—standard chow + 1% cholesterol + simvastatin 5 mg/kg b.w. Values are presented as mean ± SD. * *p* < 0.05 vs. P. ** *p* < 0.05 vs. CHOL.

**Figure 2 ijms-25-01199-f002:**
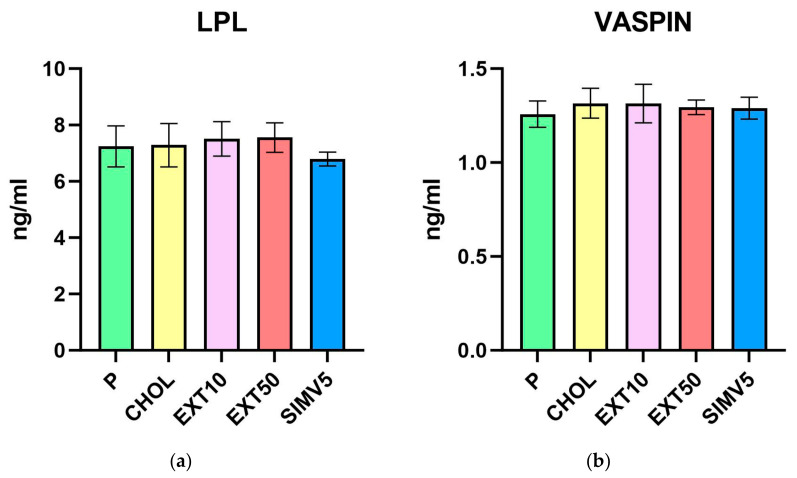
Visceral adipose tissue homogenates levels of (**a**) lipoprotein lipase (LPL) and (**b**) visceral adipose tissue-derived serine protease inhibitor (Vaspin) measured by ELISA method. P—standard chow; CHOL—standard chow + 1% cholesterol; EXT10—standard chow + 1% cholesterol + cornelian cherry extract 10 mg/kg b.w.; EXT50—standard chow + 1% cholesterol + cornelian cherry extract 50 mg/kg b.w.; SIMV5—standard chow + 1% cholesterol + simvastatin 5 mg/kg b.w. Values are presented as mean ± SD.

**Figure 3 ijms-25-01199-f003:**
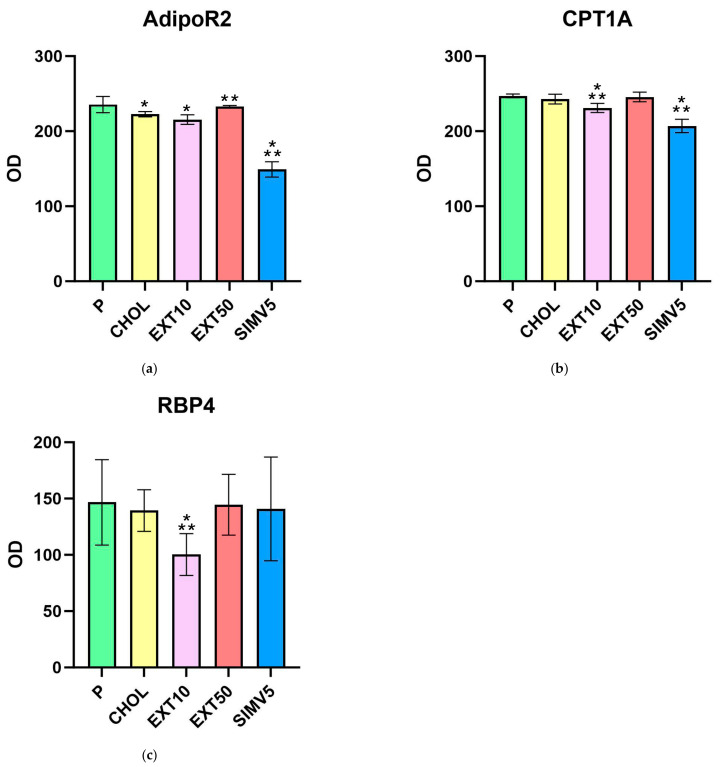
Liver homogenates levels of (**a**) adiponectin receptor 2 (AdipoR2) and (**b**) carnitine palmitoyltransferase 1A (CPT1A) assessed by Western blot method. Visceral adipose tissue homogenates levels of (**c**) retinol-binding protein 4 (RBP4) assessed by Western blot. OD—optical density. P—standard chow; CHOL—standard chow + 1% cholesterol; EXT10—standard chow + 1% cholesterol + cornelian cherry extract 10 mg/kg b.w.; EXT50—standard chow + 1% cholesterol + cornelian cherry extract 50 mg/kg b.w.; SIMV5—standard chow + 1% cholesterol + simvastatin 5 mg/kg b.w. Values are presented as mean ± SD. * *p* < 0.05 vs. P. ** *p* < 0.05 vs. CHOL.

**Figure 4 ijms-25-01199-f004:**
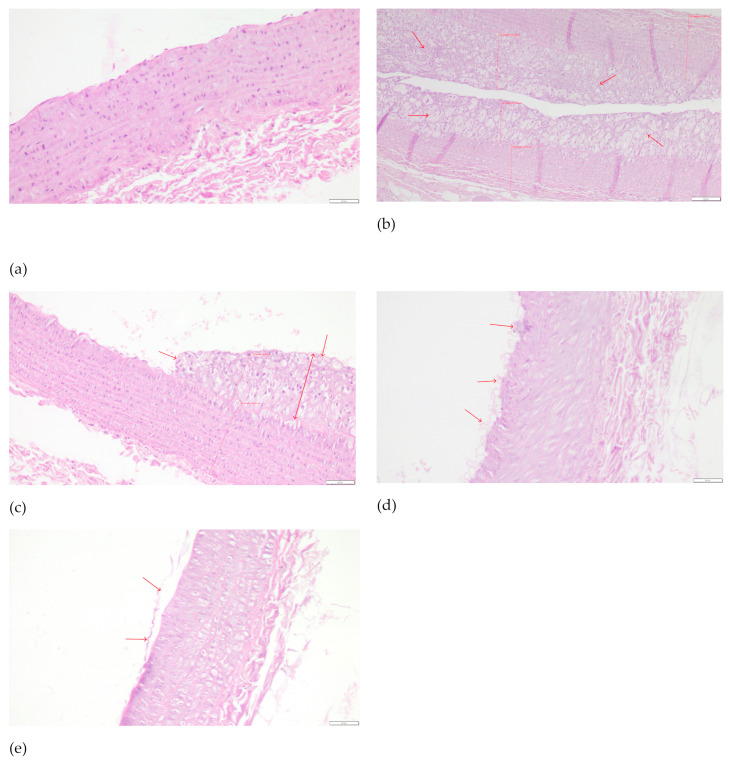
Cross-sections of the common carotid arteries segments with the most acute changes observed in the given groups, stained with hematoxylin–eosin: (**a**) P—0—no changes; magnification 200×, scale bar 50 μm. (**b**) CHOL—5—very extensive atherosclerotic plaque, often covering the entire circumference of the vessel. Red arrows indicate thick atherosclerotic plaque. Visible examples of intima and media measurement sites. Magnification 100×, scale bar 100 μm. (**c**) EXT10—3—focally located large plaque/foam cell cluster. Red arrows indicate atherosclerotic pathological changes. Visible examples of intima and media measurement sites. Magnification 200×, scale bar 50 μm. (**d**) EXT50—2—single foam cells or their small clusters on the endothelial surface. Red arrows indicate the cells on the endothelial surface. Magnification 200×, scale bar 50 μm. (**e**) SIMV5—1—slight damage of endothelial cells, no signs of atherosclerotic plaque. Red arrows indicate slight damage of endothelial surface. Magnification 200×, scale bar 50 μm.

**Figure 5 ijms-25-01199-f005:**
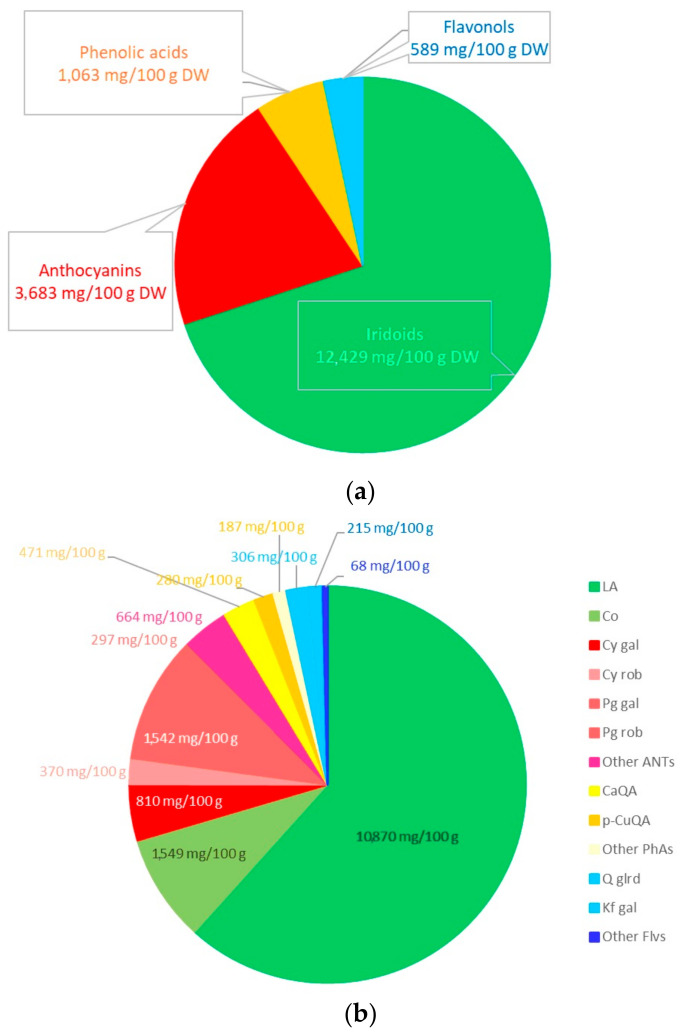
Contents (mg/100g of dry weight) of iridoids, anthocyanins, phenolic acids, and flavonols in the resin-purified cornelian cherry fruit extract (EXT) in aggregate (**a**) and detailed (**b**) presentation. Iridoids (LA—loganic acid, Co—cornuside), anthocyanins (Cy gal—cyanidin 3-O-galactoside; Cy rob—cyanidin 3-O-robinobioside; Pg gal—pelargonidin 3-O-galactoside; Pg rob—pelargonidin 3-O-robinobioside; ANTs—anthocyanins), phenolic acids (CaQA—caffeoylquinic acid; *p*-CuQA—p-coumaroilquinic acid; PhAs—phenolic acids) and flavonols (Q glrd—quercetin 3-O-glucuronide; Kf gal—kaempferol 3-O-galactoside; Flvs—flavonols).

**Figure 6 ijms-25-01199-f006:**
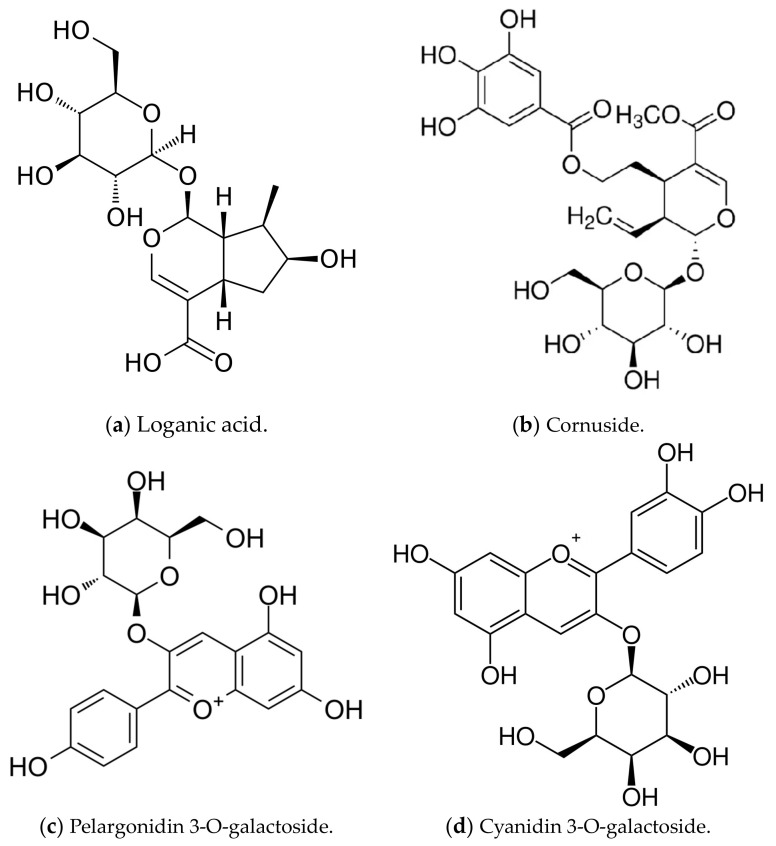
Dominant iridoid (**a**,**b**) and anthocyanin (**c**,**d**) compounds in the resin-purified cornelian cherry fruit extract (EXT).

**Table 1 ijms-25-01199-t001:** Liver homogenates levels of ATP-binding cassette protein A1 (ABCA1), ATP-binding cassette protein G1 (ABCG1), sterol regulatory element-binding protein-1c (SREBP-1c), CCAAT/enhancer binding protein alpha (C/EBPα), fatty acid synthase (FAS), and endothelial lipase (LIPG) by ELISA method. P—standard chow; CHOL—standard chow + 1% cholesterol; EXT10—standard chow + 1% cholesterol + cornelian cherry extract 10 mg/kg b.w.; EXT50—standard chow + 1% cholesterol + cornelian cherry extract 50 mg/kg b.w.; SIMV5—standard chow + 1% cholesterol + simvastatin 5 mg/kg b.w. Values are presented as mean ± SD in ng/mL or pg/mL (FAS).

Experimental Group	P	CHOL	EXT10	EXT50	SIMV5
ABCA1	2.345 ± 0.043	2.248 ± 0.109	2.343 ± 0.038	2.306 ± 0.051	2.297 ± 0.038
ABCG1	2.375 ± 0.081	2.203 ± 0.133	2.273 ± 0.103	2.376 ± 0.113	2.307 ± 0.038
SREBP-1c	1.984 ± 0.404	2.070 ± 0.379	1.536 ± 0.121	1.443 ± 0.182	1.927 ± 0.369
C/EBPα	1.685 ± 0.228	1.668 ± 0.231	1.361 ± 0.072	1.327 ± 0.105	1.631 ± 0.196
FAS	381.2 ± 6.339	375.4 ± 8.934	368.3 ± 7.088	372.3 ± 4.968	379.4 ± 8.303
LIPG	5.495 ± 0.491	5.625 ± 0.849	5.308 ± 0.409	5.484 ± 0.488	5.134 ± 0.418

**Table 2 ijms-25-01199-t002:** Visceral adipose tissue homogenates levels of lipoprotein lipase (LPL) and visceral adipose tissue-derived serine protease inhibitor (Vaspin) measured by ELISA method. P—standard chow; CHOL—standard chow + 1% cholesterol; EXT10—standard chow + 1% cholesterol + cornelian cherry extract 10 mg/kg b.w.; EXT50—standard chow + 1% cholesterol + cornelian cherry extract 50 mg/kg b.w.; SIMV5—standard chow + 1% cholesterol + simvastatin 5 mg/kg b.w. Values are presented as mean ± SD in ng/mL.

Experimental Group	P	CHOL	EXT10	EXT50	SIMV5
**LPL**	7.242 ± 0.732	7.285 ± 0.771	7.505 ± 0.613	7.552 ± 0.523	6.792 ± 0.249
**VASPIN**	1.259 ± 0.070	1.316 ± 0.079	1.314 ± 0.102	1.294 ± 0.039	1.290 ± 0.058

**Table 3 ijms-25-01199-t003:** Liver homogenates levels of adiponectin receptor 2 (AdipoR2) and carnitine palmitoyltransferase 1A (CPT1A) assessed by Western blot method. Visceral adipose tissue homogenates levels of retinol-binding protein 4 (RBP4) assessed by Western blot. P—standard chow; CHOL—standard chow + 1% cholesterol; EXT10—standard chow + 1% cholesterol + cornelian cherry extract 10 mg/kg b.w.; EXT50—standard chow + 1% cholesterol + cornelian cherry extract 50 mg/kg b.w.; SIMV5—standard chow + 1% cholesterol + simvastatin 5 mg/kg b.w. Values of optical density of the bands are presented as mean ± SD.

Experimental Group	P	CHOL	EXT10	EXT50	SIMV5
ADIPOR2	235.552 ± 10.878	222.667 ± 3.486	215.424 ± 6.400	232.876 ± 1.486	149.080 ± 10.304
CPT1A	247.143 ± 2.416	242.898 ± 6.466	230.911 ± 6.102	245.646 ± 6.383	206.956 ± 8.875
RBP4	146.672 ± 37.889	139.405 ± 18.550	100.352 ± 18.595	144.537 ± 26.933	140.804 ± 46.106

**Table 4 ijms-25-01199-t004:** Resistive index (RI) measured in common carotid arteries and aortas on day 60 (last) of the study. P—standard chow; CHOL—standard chow + 1% cholesterol; EXT10—standard chow + 1% cholesterol + cornelian cherry extract 10 mg/kg b.w.; EXT50—standard chow + 1% cholesterol + cornelian cherry extract 50 mg/kg b.w.; SIMV5—standard chow + 1% cholesterol + simvastatin 5 mg/kg b.w. Values are presented as mean ± SD.

Experimental Group	P	CHOL	EXT10	EXT50	SIMV5
CCA RI	0.685 ± 0.090	0.735 ± 0.054	0.747 ± 0.104	0.707 ± 0.093	0.741 ± 0.080
Aorta RI	0.795 ± 0.099	0.840 ± 0.084	0.763 ± 0.069	0.738 ± 0.085	0.650 ± 0.119

**Table 5 ijms-25-01199-t005:** Histopathological evaluation of the common carotid arteries. P—standard chow; CHOL—standard chow + 1% cholesterol; EXT10—standard chow + 1% cholesterol + cornelian cherry extract 10 mg/kg b.w.; EXT50—standard chow + 1% cholesterol + cornelian cherry extract 50 mg/kg b.w.; SIMV5—standard chow + 1% cholesterol + simvastatin 5 mg/kg b.w. Values are presented as mean ± SD. 0—no changes; 1—slight damage of endothelial cells, no signs of atherosclerotic plaque; 2—single foam cells or their small clusters on the endothelial surface; 3—focally located large plaque/foam cell cluster; 4—large focus or several individual clusters of foam cells; 5—very extensive atherosclerotic plaque, often covering the entire circumference of the vessel.

Experimental Group	P	CHOL	EXT10	EXT50	SIMV5
CCA Evaluation	0.444 ± 0.527	3.625 ± 1.408	0.700 ± 1.059	0.556 ± 0.726	0.222 ± 0.441

**Table 6 ijms-25-01199-t006:** Experimental groups and feeding schema of the study.

Experimental Group	P	CHOL	EXT10	EXT50	SIMV5
Chow	standard chow (sc)	sc + 1% cholesterol	sc + 1% cholesterol	sc + 1% cholesterol	sc + 1% cholesterol
Tested substance	- (normal saline solution)	- (normal saline solution)	cornelian cherry extract 10 mg/kg/b.w.	cornelian cherry extract 50 mg/kg/b.w.	simvastatin 5 mg/kg/b.w.

## Data Availability

The data underlying this article will be shared upon request with the corresponding authors.

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
