# Peer review of "Cornelian Cherry (Cornus mas L.) Fruit Extract Lowers SREBP-1c and C/EBPα in Liver and Alters Various PPAR-α, PPAR-γ, LXR-α Target Genes in Cholesterol-Rich Diet Rabbit Model"

_ijms, 2024, doi:10.3390/ijms25021199_

Round 1
Reviewer 1 Report
Comments and Suggestions for Authors
Major comments
1. The manuscript is excessively long with few results presented in the manuscript. Sentences in the manuscript are very long, making it difficult for viewers to comprehend
2. In discussing the results, the author focus more on the results of their previous findings and scientific literature that the actual results obtained in the study thus making the discussion section to be unnecessary long
3. A major flaw in the study is that the 1% chol in the diet was not able to induce metabolic changes in the manuscript thereby making it difficult to assess with confidence the effects of the fruit extract
4. There are a lot of grammatical errors in the manuscript arising from the long sentences. Also, I recommend a scientific English editor to edit the manuscript. A lot of woods used in the manuscript are not scientific. Arthurs should use scientific words in their manuscript. E.g, mitigation in lines 113
5. It will be good for the authors to present the representative pictures of the western blot bands in the manuscript or the pictures can be placed in a supplementary section
6. Picture quality in figure 4 is very poor and also poorly presented. What pathological changes can be observed on the stained tissue? This is not explained in the results. Arthurs should present the pathological changes with the help of arrows
Minor comments:
1. Abbreviations have been poorly used throughout the manuscripts. For example, in the abstract and in figure legends, abbreviations are used without their full meaning.
2. Authors should avoid presenting results in the introduction section of the manuscript. They may highlight result conclusion but not present the results as in the case of the manuscript
3. Most of the information written in the introduction could have been discussed in the discussion section. E.g lines 112-128
4. It will be good for the authors to state the exact levels of the various parameters measured in the study (where applicable) and not state just the P values. This makes it difficult for authors to appreciate the changes in the levels of the parameters used in the study e.g, lines 147-149. Authors should also state whether the levels of the parameters were increased or decreased in relation to the P and CHOL group. E.g, 153-156
5. ‘**’ is described in the figure legend but not in the figure. *** is described in the figure but not in the figure legend
6. In lines 189-190, Arthur stated that the ADIPO2 was decreased in the EXT50 group, but looking at the figure, there seems to be no difference with the CHOl group.
7. It will be a major flaw for Arthurs to conclude base on the findings of this study that irridoid and anthocyanin were the main bio-actives of the extract. Their abundance does not necessarily mean they were the main active substance
8. How was the 1% Cholesterol administered to the rabbit? Or how was the chow enriched with 1% Cholesterol
9. Figure 5 can be move to result section
10. Materials used is recommended to be listed in a separate section in the material and method section to make the material and method section less complex
11. The conclusion section can be shorten and just summarize the key findings in the manuscript especially as the discussion section is too long and complex
Comments on the Quality of English Language1. There are a lot of grammatical errors in the manuscript arising from the long sentences. Also, I recommend a scientific English editor to edit the manuscript. A lot of woods used in the manuscript are not scientific. Arthurs should use scientific words in their manuscript. E.g, mitigation in lines 113
Author Response
Respected Reviewer,
On behalf of the co-authors and myself, I would like to thank you for your time devoted to the thorough study of the content of our manuscript and all valuable hints and accurate comments. We are confident that they have helped us improve the manuscript markedly.
We provide our response to each suggestion below:
Major comments:
- The manuscript is excessively long with few results presented in the manuscript. Sentences in the manuscript are very long, making it difficult for viewers to comprehend.
Thank you for pointing this out. The main author has this writing style and tries to shorten and simplify sentences, although he is not always successful. The article is slightly longer in content because it is the end of a multi-year project and constitutes its summary (hence the references to several previous publications in the text).
- In discussing the results, the author focus more on the results of their previous findings and scientific literature that the actual results obtained in the study thus making the discussion section to be unnecessary long.
In our opinion, the current results are described appropriately, following the scheme: what result was obtained, whether similar studies were carried out, and, if so, how their results correspond to ours, and then an explanation or speculations about what such result may mean in general terms. As mentioned, this article summarizes a large project, hence the references to previous publications.
- A major flaw in the study is that the 1% chol in the diet was not able to induce metabolic changes in the manuscript thereby making it difficult to assess with confidence the effects of the fruit extract.
1% cholesterol can induce metabolic changes in the rabbit model, as evidenced by changes in the parameters described in the article (including the previous ones). Sometimes, in the discussion, we only refer to the fact that it is difficult to interpret the results concerning a long-term period. The model used in the study is a classic, recognized, and widely used research model, see: Niimi M. et al. “Hyperlipidemic Rabbit Models for Anti-Atherosclerotic Drug Development” in Applied Sciences 2020; and Yanni AE. “The laboratory rabbit: an animal model of atherosclerosis research” in Lab Anim. 2004.
- There are a lot of grammatical errors in the manuscript arising from the long sentences. Also, I recommend a scientific English editor to edit the manuscript. A lot of woods used in the manuscript are not scientific. Arthurs should use scientific words in their manuscript. E.g, mitigation in lines 113.
The article was double-checked for linguistic correctness by a colleague fluent in English and the Grammarly application, and appropriate changes were made.
- It will be good for the authors to present the representative pictures of the western blot bands in the manuscript or the pictures can be placed in a supplementary section.
The Western Blot files have already been uploaded to the submission system as unpublished materials, in consultation with editor Ms. Winifred Wang. Thank you.
- Picture quality in figure 4 is very poor and also poorly presented. What pathological changes can be observed on the stained tissue? This is not explained in the results. Arthurs should present the pathological changes with the help of arrows.
Thank you for pointing this out. The figures have been changed, and they are brighter and more precise. The arrows show the observed changes. With such small pictures, minor changes are not very visible.
Minor comments:
- Abbreviations have been poorly used throughout the manuscripts. For example, in the abstract and in figure legends, abbreviations are used without their full meaning.
Revised, thank you.
- Authors should avoid presenting results in the introduction section of the manuscript. They may highlight result conclusion but not present the results as in the case of the manuscript.
The results presented in the introduction concern previously published works - their presentation is necessary because it justifies why specific parameters were selected for determination in this study and provides a context for reference in the discussion.
- Most of the information written in the introduction could have been discussed in the discussion section. E.g lines 112-128.
As above.
4. It will be good for the authors to state the exact levels of the various parameters measured in the study (where applicable) and not state just the P values. This makes it difficult for authors to appreciate the changes in the levels of the parameters used in the study e.g, lines 147-149. Authors should also state whether the levels of the parameters were increased or decreased in relation to the P and CHOL group. E.g, 153-156. The results are presented in the added tables. Lines 153-156 have been corrected. Thank you.
- ‘**’ is described in the figure legend but not in the figure. *** is described in the figure but not in the figure legend.
clusions have been shortened.jrzysty, pozwalający szybko znaleźć odpowiednią informację o danym materiale w subsekcji po prosThere are not ***. There are only * p < 0.05 vs. P. and ** p < 0.05 vs. CHOL. If there are three asterisks next to a given bar in the figure, they do not mean a different comparison but that there was a difference between both the P and CHOL groups. That is why one and two asterisks are not on one line (which might suggest three asterisks) but on two lines below each other.
- In lines 189-190, Arthur stated that the ADIPO2 was decreased in the EXT50 group, but looking at the figure, there seems to be no difference with the CHOl group.
It was not a depletion (as written in the text) but an increase. Thank you for your vigilance and catching such a significant distortion.
- It will be a major flaw for Arthurs to conclude base on the findings of this study that irridoid and anthocyanin were the main bio-actives of the extract. Their abundance does not necessarily mean they were the main active substance.
We do not conclude from this study that iridoids and anthocyanins were the main active substances of the extract. This statement follows from many other reports on cornelian cherry fruits and their extract. Citations to several such articles, including a review article containing references to others, are included in the introduction.
8. How was the 1% Cholesterol administered to the rabbit? Or how was the chow enriched with 1% Cholesterol.
Standard chow was mixed with cholesterol in a weight ratio of 99:1.
- Figure 5 can be move to result section.
An interesting concept, thank you for that. However, the cornelian cherry extract was used to observe the changes under its influence in the rabbit model. Therefore, it is the material to obtain results, and the determined composition was not the result itself. To avoid distracting readers from the observed changes in the parameters described, we prefer that Figure 5 remains in the materials and methods section.
- Materials used is recommended to be listed in a separate section in the material and method section to make the material and method section less complex.
Materials used for individual determinations are described in subsections describing the given determination. Such a provision seems transparent. It allows quickly finding appropriate information about a given material simply in the subsection describing a given test.
- The conclusion section can be shorten and just summarize the key findings in the manuscript especially as the discussion section is too long and complex.
Conclusions have been shortened.
We hope you will be satisfied with revisions and re-consider our manuscript for publication in the IJMS journal. Thank you very much.
Best regards
Maciej Danielewski

Reviewer 2 Report
Comments and Suggestions for Authors
General comments:
The manuscript is well written. The authors investigated “Cornelian Cherry (Cornus mas L.) Fruit Extract Lowers 2 SREBP-1c and C/EBPα in Liver and Alters Various PPAR-α, 3 PPAR-γ, LXR-α Target Genes in Cholesterol-Rich Diet Rabbit 4 Model”. The study is interesting and adds to the existing body of knowledge. The data and results of the current study are well presented and the findings are beneficial to the pharmaceutical industry. The current study has shown that. However, there are a few things that need clarification and revisions.
Details comments:
1. Page 1, Abstract Lines 35-41: Please simplify the objectives of the study. The objectives of the study should be the methodology in this section.
2. Some of the abbreviations in the abstract should be written in full name and then abbreviated.
3. Page 2, Lines 54-58: Please add the latest prevalence (2020-2023) of non-communicable diseases (obesity/cardiovascular) in this section.
4. Page 2, Lines 71-72: Please write dosage in the unit. 10 mg/kg and 50 mg/kg.
5. Page 3, Line 125: I/M is intima/media ratio?
6. Page 5, Lines 170-176: The authors should add the full name for each abbreviation in Figure 2. Authors should revise other Figure legends (Figures 2 and 3).
7. Page 9, Figure 4: Authors should add the thickness of I/M and label the tissue with arrows. Please replace the histopathological images with high-resolution images. Current images are unclear and difficult for the reviewer to differentiate the cells, hemorrhage, etc. Please revise.
8. Discussion: No issue was found. The authors discuss the current findings.
9. Page 15, Line 570: New Zealand white rabbit. Please add the body weight range.
10. Page 15, Line 580: 10 mg/kg
11. Page 16, Line 583: How were the rabbits were anesthetized? Using drugs or cervical dislocation or that procedure? Please revise this section and add the current citations.
12. Page 18, Lines 635-656: Please add the citations for western blot protocols.
13. Page 19, Lines 674-75: Please add the complete protocols for H&E staining with citations. Please add the citations for scoring for this section.
14. Conclusion: Please revise the conclusion section and add the major findings of the current study. The current conclusion is too long and should be simplify into one paragraph.
Comments on the Quality of English Language
The English language is fine and minor editing is required.
Author Response
Respected Reviewer,
On behalf of the co-authors and myself, I would like to thank you for your time devoted to the thorough study of the content of our manuscript and all valuable hints and accurate comments. We are confident that they have helped us improve the manuscript markedly.
We provide our response to each suggestion below:
- Page 1, Abstract Lines 35-41: Please simplify the objectives of the study. The objectives of the study should be the methodology in this section.
Thank you for pointing this out. We have rewritten this section of abstract.
- Some of the abbreviations in the abstract should be written in full name and then abbreviated.
Done, thank you.
- Page 2, Lines 54-58: Please add the latest prevalence (2020-2023) of non-communicable diseases (obesity/cardiovascular) in this section.
Done, thank you.
- Page 2, Lines 71-72: Please write dosage in the unit. 10 mg/kg and 50 mg/kg.
Done, thank you.
- Page 3, Line 125: I/M is intima/media ratio?
Yes, full terms added.
- Page 5, Lines 170-176: The authors should add the full name for each abbreviation in Figure 2. Authors should revise other Figure legends (Figures 2 and 3).
Thank you for your vigilance. Full names added.
- Page 9, Figure 4: Authors should add the thickness of I/M and label the tissue with arrows. Please replace the histopathological images with high-resolution images. Current images are unclear and difficult for the reviewer to differentiate the cells, hemorrhage, etc. Please revise.
Thank you for pointing this out. The figures show example measurement locations. The photos were changed to be brighter and more precise. In such small figures, cells are less visible. You can see atherosclerotic plaque of varying thickness or damage to the endothelium. No hemorrhages are shown in the photos.
- Discussion: No issue was found. The authors discuss the current findings.
Thank you.
- Page 15, Line 570: New Zealand white rabbit. Please add the body weight range.
Done, thank you.
- Page 15, Line 580: 10 mg/kg
Done, thank you.
- Page 16, Line 583: How were the rabbits were anesthetized? Using drugs or cervical dislocation or that procedure? Please revise this section and add the current citations.
Rabbits were anesthetized using drugs. Proper information added. Thank you.
- Page 18, Lines 635-656: Please add the citations for western blot protocols.
Done, thank you.
- Page 19, Lines 674-75: Please add the complete protocols for H&E staining with citations. Please add the citations for scoring for this section.
HE staining protocol inserted in the Histopathological Evaluation of the Common Carotid Arteries section, reference inserted in the References section (item 141)
- Conclusion: Please revise the conclusion section and add the major findings of the current study. The current conclusion is too long and should be simplify into one paragraph.
Although we did not manage to fit into one paragraph, we believe that we have significantly shortened the conclusions, making them easier to comprehend.
We hope you will be satisfied with revisions and re-consider our manuscript for publication in the IJMS journal. Thank you very much.
Best regards
Maciej Danielewski

Round 2
Reviewer 1 Report
Comments and Suggestions for Authors
The authors have responded accurately to my comments on the manuscript. I recommend the manuscript to be published in its present format.
Reviewer 2 Report
Comments and Suggestions for Authors
General Comments
The manuscript “Cornelian Cherry (Cornus mas L.) Fruit Extract Lowers 2 SREBP-1c and C/EBPα in Liver and Alters Various PPAR-α, 3 PPAR-γ, LXR-α Target Genes in Cholesterol-Rich Diet Rabbit 4 Model” is significantly improved after the authors' corrections. The authors have appropriately responded to the original concerns. The figures for data in the manuscript have been replaced as suggested. I think the changes are acceptable. I recommend the publication of the manuscript.